# Association between prediabetes and depression: A meta-analysis

**Yi Yu**[1‡], **Weitao Wan**[2‡*]

1 Department of Psychiatry, Wuchang Hospital Affiliated to Wuhan University of Science and Technology, Wuhan, Hubei, China, 2 Department of Psychiatry, Tianyou Hospital Affiliated to Wuhan University of Science and Technology, Wuhan, Hubei, China

‡ YY and WW are contributed equally to this work and share first authorship on this work.
* wwt686@sohu.com

**Data Availability Statement:** The authors confirm that the data supporting the findings of this study are available within the article.

**Funding:** The author(s) received no specific funding for this work.

## Abstract

### Background

Previous studies evaluating the association between prediabetes and depression have shown inconsistent results. Consequently, the aim of the systematic review and meta-analysis was to investigate whether prediabetes is associated with depression in the general population.

### Methods

Relevant observational studies were obtained by searching the Medline, Web of Science, and Embase databases. A random-effects model was utilized to pool the results by incorporating the influence of heterogeneity. Multiple subgroup analysis was performed to evaluate the influence of the study characteristics on the outcome.

### Results

Sixteen large-scale cross-sectional studies involving 322,863 participants were included. Among the total participants, 82,154 (25.4%) had prediabetes. The pooled results showed that prediabetes was associated with a higher prevalence of depression in this population (odds ratio [OR]: 1.16, 95% confidence interval [CI]: 1.05 to 1.28, $p = 0.003$; $I^2 = 58\%$). Subgroup analysis showed a stronger association between prediabetes and depression in younger subjects (<50 years old, OR: 1.25, 95% CI: 1.04 to 1.50) than that in older subjects (≥50 years old, OR: 1.05, 95% CI: 1.10 to 1.10; $p$ for subgroup difference = 0.03). Other study characteristics, such as the study country, sex of the participants, definition of prediabetes, methods for the detection of depression, and study quality score, did not seem to significantly affect the results ($p$ for subgroup difference all > 0.05).

### Conclusions

Prediabetes may be associated with a slightly higher prevalence of depression in the general population, particularly in subjects aged <50 years old.

**Competing interests:** The authors have declared that no competing interests exist.

## Introduction

Depression is a common affective disorder in patients with chronic diseases [1–3]. As a common metabolic disorder, the prevalence of diabetes has increased in recent decades in both developing and developed countries [4]. Accumulating evidence suggests that people with diabetes have an increased risk of depression [5, 6]. In a previous meta-analysis, it was shown that approximately more than half of patients with diabetes suffer from depression [7]. In children and adolescents with type 1 diabetes, the pooled prevalence of depression was reported to be nearly 30% [8]. Interestingly, subsequent studies suggest that the association between diabetes and depression seems to be bidirectional [9–11]. Besides a high prevalence of depression in patients with diabetes, it is also shown that various measures of depression could be used to predict the risk of type 2 diabetes, such as depression as evidenced by symptom scales, patient diagnosis, face-to-face interviews, and the use of antidepressants [12]. With the recent research advances in the field of diabetology, an intermediate state of hyperglycemia between normoglycemia and diabetes has been proposed, which is termed as prediabetes [13, 14]. Clinically, prediabetes refers to the status of impaired glucose regulation before a diagnosis of diabetes, which includes impaired fasting glucose (IFG), impaired glucose tolerance (IGT), and mildly elevated glycolated hemoglobin (HbA1c: 5.7% to 6.4%) [15]. Similar to diabetes, people with prediabetes have also been associated with an increased risk of cardiovascular diseases [16]. In view of the close relationship between diabetes and depression, it is interesting to determine if prediabetes is also associated with depression in the general population [17]. However, the results of previous studies were not consistent [18–33]. Some of them supported that prediabetes was related to depression [24–27, 29, 30], while other studies did not found a statistically significant association [18–23, 28, 31–33]. In addition, these studies are with populations from various places and of different study definitions and methodologies for evaluating prediabetes and depression [18–33]. It remains unknown whether these factors may influence the association between prediabetes and depression. Consequently, in this study, we performed a systematic review and meta-analysis to investigate whether prediabetes is associated with depression in the general population.

## Methods

The Preferred Reporting Items for Systematic Reviews and Meta-Analyses (PRISMA 2020) [34, 35] and the Cochrane Handbook for Systematic Reviews and Meta-analyses [36] were followed in this meta-analysis during the study design, data collection, statistical analysis, and results interpretation.

### Literature search

To identify studies relevant to the aim of the meta-analysis, we searched the Medline, Web of Science, and Embase databases utilizing comprehensive search terms involving: (1) "prediabetes" OR "pre-diabetes" OR "prediabetic" OR "pre-diabetic" OR "prediabetic state" OR "borderline diabetes" OR "impaired fasting glucose" OR "impaired glucose tolerance" OR "IFG" OR "IGT"; and (2) "depression" OR "depressive". The search terms were based on key words rather than MeSH terms to improve the sensitivity of the database search. However, a comparison with MeSH terms was performed before database search to ensure all relevant MeSH terms are included in the search terms.

### Inclusion and exclusion criteria

The inclusion criteria for the potential studies were: (1) large-scale observational studies published as full-length articles (sample size ≥ 1000); (2) studies conducted in adults (18 years and

older); (3) prediabetes and depression were evaluated with the same methods and diagnostic criteria in accordance with those used in the original studies; and (4) reported the prevalence/incidence of depression compared between participants with prediabetes versus normoglycemia in a multivariate analysis, as well at least adjusting for age and sex.

The exclusion criteria were: (1) small-scale studies, studies including patients who were diagnosed with specific diseases rather than a general population, or studies with univariate analysis; (2) studies did not evaluate prediabetes or did not report depression; or (3) preclinical studies, reviews, or editorials. If studies with overlapping populations were retrieved, the one with the largest sample size was included for the meta-analysis.

The search was limited to studies in humans. Also, we only considered studies published as full-length articles in peer-reviewed journals in English. In addition, the references of related original and review articles were also manually screened for identifying potentially related studies. The literatures published from the inception of the databases to December 8, 2023 were screened.

## Study quality evaluation and data extraction

The processes for the literature search, study identification, study quality evaluation, and data collection were independently conducted by two authors. In case of disagreement, the two authors discussed it to reach a consensus. We used the Newcastle–Ottawa Scale (NOS) [37] for assessing the quality of the included studies. This scale consisted of three aspects, including selection of the population, control of confounders, and outcome measurement and analysis. The total NOS scores ranged from 1 to 9, with 9 indicating the best quality. The following data were extracted from each study for subsequent analysis: the study information (author, year, country, and design), participant characteristics (sample size, age, and sex), diagnosis of prediabetes (definition and number of participants with prediabetes), diagnosis of depression (methods and number of participants with depression), and variables adjusted when the association between prediabetes and depression was reported.

## Statistics

The association between prediabetes and depression was summarized as the odds ratio (OR) and corresponding 95% confidence interval (CI). By using 95% CIs or $p$-values, the ORs and standard errors (SEs) could be calculated, while a subsequent logarithmical transformation kept the variance stabilized and normalized. The Cochrane Q test and $I^2$ statistics were used to estimate study heterogeneity [38], with significant heterogeneity reflected by $I^2 > 50\%$. The results were combined using a random-effects model incorporating heterogeneity's influence [36]. Sensitivity analyses by omitting one study at a time were performed to investigate the robustness of the findings. Predefined subgroup analyses were also performed to evaluate the influences of the study characteristics on the outcome. The medians of the continuous variables were used as the cutoffs for defining the subgroups. The estimation of publication bias underlying the meta-analysis was first achieved by construction of funnel plots and visual inspection of the plot symmetry [39]. An Egger's regression test was also performed [39]. The statistical analysis was carried out using RevMan (Version 5.1; Cochrane Collaboration, Oxford, UK) and Stata software (version 12.0; Stata Corporation, College Station, TX). A two-sided $p < 0.05$ suggested statistical significance.

## Results

### Study inclusion

The process for identifying relevant studies for study inclusion in the meta-analysis is presented in Fig 1. In brief, 2032 potentially relevant records were obtained after comprehensive

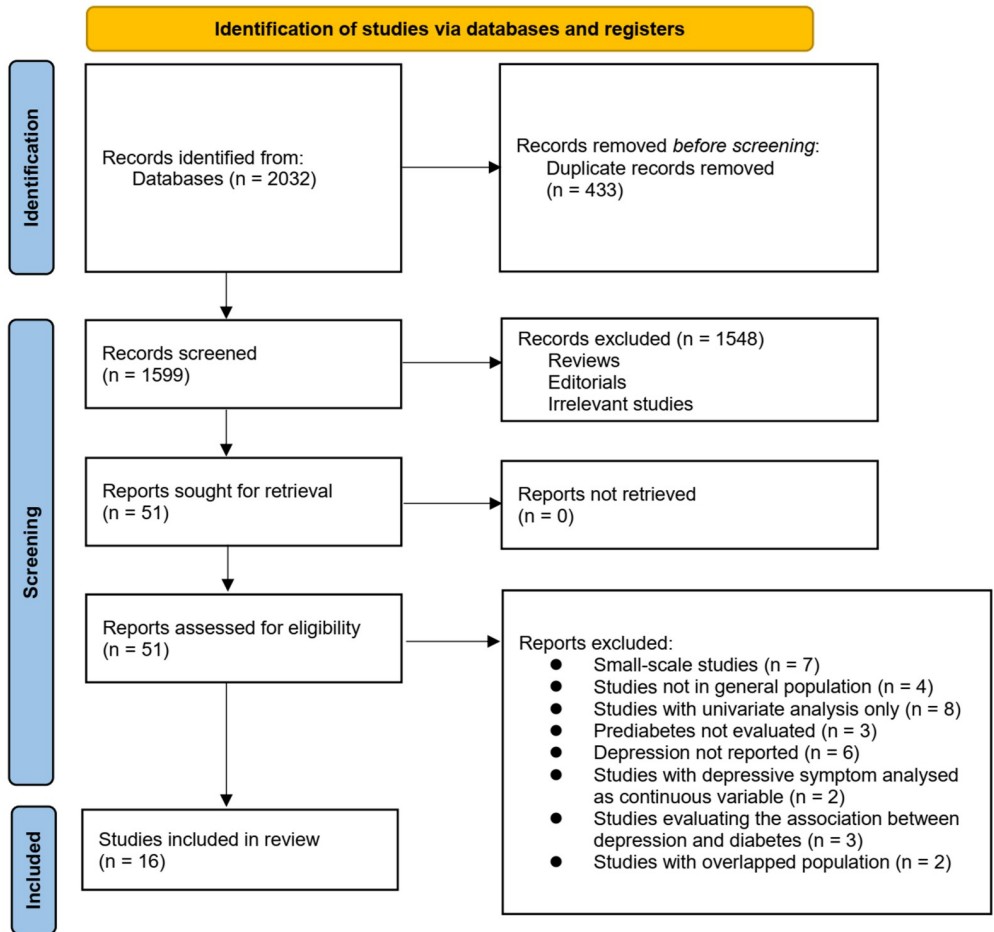

**Fig 1. Flowchart depicting the database search and study inclusion processes.**

searches of the three databases, with 433 studies then excluded due to duplication. Subsequently, a screening via considering the titles and abstracts of the remaining records led to the exclusion of a further 1548 studies, mostly because they were not related to the aim of the meta-analysis, leaving 51 studies remaining. Accordingly, the full texts of the remaining 51 studies were read by two independent authors, and 35 of them were further removed for various reasons, as listed in Fig 1. Finally, 16 observational studies remained and were considered to be suitable for the subsequent quantitative analyses [18–33].

## Overview of the study characteristics

Table 1 presents the summarized characteristics of the included studies. Overall, 16 cross-sectional studies [18–33] were included in the meta-analysis. These studies were reported from 2007 to 2023, and performed in Finland, the Netherlands, the United Kingdom, the United States, India, China, Bangladesh, Austria, and Korea. All of the studies had large sample sizes, ranging from 1,728 to 229,047. Community-derived general populations were included in all of the included studies except for one study, which enrolled US veterans [23]. The mean ages of the participants were 38.1 to 66.1 years old. As for the definition of prediabetes, IFG was used in three studies [18, 23, 24], IFG and/or IGT were used in ten studies [19–22, 25–27, 30–

**Table 1. Characteristics of the included studies.**

| Study | Country | Design | Population | Sample size | Mean age | Male | Definition of PreD | No. of people with PreD | Diagnosis for depression | No. of subjects with depression | Variables adjusted |
|---|---|---|---|---|---|---|---|---|---|---|---|
| | | | | | years | % | | | | | |
| Paile 2007 | Finland | CS | Community-derived population | 2003 | 61.5 | 46.3 | IFG and/or IGT | 635 | BDI | 383 | Age, sex, BMI, and prevalence of CVD |
| Knol 2007 | The Netherlands | CS | Community-derived population | 4747 | 39.4 | 46.7 | IFG | 671 | SCL-90 | 916 | Age, sex, education, BMI, smoking, alcohol drinking, physical exercise, and number of chronic diseases |
| Holt 2009 | UK | CS | Community-derived population | 2995 | 66.1 | 52.7 | IFG and/or IGT | 996 | HAD-D | 161 | Age, sex, BMI, smoking, social class and alcohol consumption |
| Aujla 2009 | UK | CS | Community-derived population | 6009 | 58 | 47.4 | IFG and/or IGT | 855 | WHO-5 | 1231 | Age, sex, BMI, smoking, WC, exercise, and Index of Multiple Deprivation score |
| Gale 2010 | The US | CS | US veterans | 4293 | 38.8 | 100 | IFG | 492 | MMPI | 276 | Age, ethnicity, clinical characteristics, and health behaviors, intelligence, educational attainment, and household income |
| Poongothai 2010 | India | CS | Community-derived population | 23787 | 38.1 | 49.3 | IFG | 7657 | PHQ-9 | 3391 | Age, sex, BMI, hypertension, and SES |
| Bouwman 2010 | The Netherlands | CS | Community-derived population | 2667 | 53.4 | 47.3 | IFG and/or IGT | 425 | CES-D | 348 | Age, sex, education, family history of DM, TG, HDL-C, TC, hypertension, smoking, and WC |
| Tsai 2012 | China | CS | Community-derived population | 9561 | 46.3 | 61.1 | IFG and/or IGT | 2440 | BSRS-50 | NR | Age, sex, BMI, marital status, educational level, hypertension, SCr, TG, and HDL-C, current smoking and alcohol use, regular exercise, and family history of DM |
| Sun 2015 | China | CS | Community-derived population | 229047 | 57.4 | 34.4 | IFG and/or IGT | 59512 | PHQ-9 | 10994 | Age, sex, BMI, HbA1c, physical activity, smoking and alcohol drinking status, education level, occupation and marital status |
| Natasha 2015 | Bangladesh | CS | Community-derived population | 2293 | 41.8 | 36.7 | IFG and/or IGT | 197 | MARDS | 351 | Age, sex, marital status, BMI, waist to hip ratio, physical activity, and hypertension |
| Albertorio 2017 | The US | CS | Community-derived population | 7717 | 53.5 | 49.3 | IFG or HbA1c (5.7~6.4%) | 2024 | PHQ-9 | 216 | Age, sex education, race-ethnicity, poverty, and BMI |
| Breyer 2019 | Austria | CS | Community-derived population | 11014 | 44.9 | 47.8 | IFG or HbA1c (5.7~6.4%) | 2225 | HAD-D | NR | Age, sex, smoking, LDL-C, HDL-C, and TG |
| Xu 2021 | China | CS | Community-derived population | 1728 | 40.1 | 38.4 | IFG and/or IGT | 536 | PHQ-9 | 83 | Age, sex, BMI, physical activity at work, and systolic pressure |
| Cui 2021 | China | CS | Community-derived population | 3300 | 40.6 | 40.2 | IFG and/or IGT | 771 | ZSDS | 179 | Age, sex, area, education, marriage, monthly income, occupation |

(*Continued*)

**Table 1.** (Continued)

| Study | Country | Design | Population | Sample size | Mean age | Male | Definition of PreD | No. of people with PreD | Diagnosis for depression | No. of subjects with depression | Variables adjusted |
|---|---|---|---|---|---|---|---|---|---|---|---|
| | | | | | years | % | | | | | |
| Yang 2023 | Korea | CS | Community-derived population | 4063 | 52 | 0 | IFG or HbA1c (5.7~6.4%) | 1577 | PHQ-9 | 261 | Age, SES, alcohol drinking, smoking, BMI, menopausal status |
| de Ritter 2023 | The Netherlands | CS | Community-derived population | 7639 | 58.8 | 50 | IFG and/or IGT | 1141 | PHQ-9 | 328 | Age, sex, education, alcohol use, smoking, BMI, physical activity, and healthy diet score, total cholesterol-to-HDL cholesterol ratio, systolic BP, and medications |

PreD, prediabetes; CS, cross-sectional; IFG, impaired fasting glucose; IGT, impaired glucose tolerance; HbA1c, glycolated hemoglobin; BDI, Beck Depression Inventory; SCL-90, Symptom Check List 90; HAD-D, Hospital Anxiety and Depression–Depressive symptoms; WHO-5, World Health Organization Five Wellbeing Index 5; MMPI, Minnesota Multiphasic Personality Inventory; PHQ-9, Patient Health Questionnaire 9 items; CES-D, Centre for Epidemiologic Studies Depression Scale; BSRS-50, Questionnaire of Brief Symptoms Rating Scale; MARDS, Montgomery Asberg Depression Rating Scale; CIDI, Composite International Diagnostic Interview; ZSDS, Zung self-rating depression scale; BMI, body mass index; CVD, cardiovascular disease; WC, waist circumference; SES, socioeconomic status; DM, diabetes mellitus; TG, triglyceride; HDL-C, high-density lipoprotein cholesterol; TC, total cholesterol; SCr, serum creatinine; LDL-C, low-density lipoprotein cholesterol; BP, blood pressure.

32], and IFG and/or increased HbA1c (5.7~6.4%) were used in the other three studies [28, 29, 33]. Accordingly, 82,154 (25.4%) of the included subjects had prediabetes. Various scales were used to identify people with depression, with the Patient Health Questionnaire 9 items (PHQ-9) the most commonly used, being applied in six studies [24, 27, 28, 31–33]. Multivariate analyses were used among all the included studies when the association between prediabetes and depression was reported, which at least accounted for potential confounding factors, such as age and sex. The NOS of the included studies were seven to nine stars, suggesting an overall good study quality (Table 2).

## Results of the meta-analysis

Since four of the included studies reported the outcome in men and women separately [21, 22, 30, 32], these datasets were independently included, which meant there were 20 datasets from 16 studies available for the meta-analysis. The pooled results with a random-effects model showed that prediabetes was associated with a higher prevalence of depression in this population (OR: 1.16, 95% CI: 1.05 to 1.28, $p$ = 0.003; $I^2$ = 58%; Fig 2).

Further sensitivity analysis performed by excluding one study at a time showed consistent results (OR: 1.12 to 1.18, $p$ all < 0.05). Further subgroup analysis according to the study country showed similar results ($p$ for subgroup analysis = 0.46; Fig 3A). The subgroup analysis showed a stronger association between prediabetes and depression in younger subjects (<50 years old, OR: 1.25, 95% CI: 1.04 to 1.50) than that in older subjects (≥50 years old, OR: 1.05, 95% CI: 1.10 to 1.10; $p$ for subgroup difference = 0.03; Fig 3B). Other study characteristics, such as the sex of the participants ($p$ for subgroup analysis = 0.44; Fig 4A), definition of prediabetes ($p$ for subgroup analysis = 0.70; Fig 4B), methods used for the detection of depression ($p$ for subgroup analysis = 0.13; Fig 5A), and study quality score ($p$ for subgroup analysis = 0.55; Fig 5B), did not seem to significantly affect the results.

**Table 2. Study quality evaluation via the Newcastle–Ottawa Scale.**

| Studies | Representativeness of the sample | Sample size | Non-responders | Ascertainment of exposure | Control for age and sex | Control for other confounding factors | Independent assessment of the outcome | Self-report outcome | Statistics reported | Total |
|---|---|---|---|---|---|---|---|---|---|---|
| Paile 2007 | 1 | 1 | 1 | 1 | 1 | 1 | 1 | 1 | 1 | 9 |
| Knol 2007 | 0 | 1 | 1 | 1 | 1 | 1 | 1 | 0 | 1 | 7 |
| Holt 2009 | 0 | 1 | 1 | 1 | 1 | 1 | 1 | 1 | 1 | 8 |
| Aujla 2009 | 0 | 1 | 1 | 1 | 1 | 1 | 1 | 0 | 1 | 7 |
| Gale 2010 | 1 | 1 | 1 | 1 | 1 | 1 | 1 | 1 | 1 | 9 |
| Poongothai 2010 | 1 | 1 | 1 | 1 | 1 | 1 | 1 | 1 | 1 | 9 |
| Bouwman 2010 | 1 | 1 | 1 | 1 | 1 | 1 | 1 | 1 | 1 | 9 |
| Tsai 2012 | 0 | 1 | 1 | 1 | 1 | 1 | 1 | 0 | 1 | 7 |
| Sun 2015 | 1 | 1 | 1 | 1 | 1 | 1 | 1 | 1 | 1 | 9 |
| Natasha 2015 | 1 | 1 | 1 | 1 | 1 | 1 | 1 | 1 | 1 | 9 |
| Albertorio 2017 | 0 | 1 | 1 | 1 | 1 | 1 | 1 | 1 | 1 | 8 |
| Breyer 2019 | 1 | 1 | 1 | 1 | 1 | 1 | 1 | 1 | 1 | 9 |
| Xu 2021 | 0 | 1 | 1 | 1 | 1 | 1 | 1 | 1 | 1 | 8 |
| Cui 2021 | 0 | 1 | 1 | 1 | 1 | 1 | 1 | 1 | 1 | 8 |
| Yang 2023 | 1 | 1 | 1 | 1 | 1 | 1 | 1 | 1 | 1 | 9 |
| de Ritter 2023 | 0 | 1 | 1 | 1 | 1 | 1 | 1 | 1 | 1 | 8 |

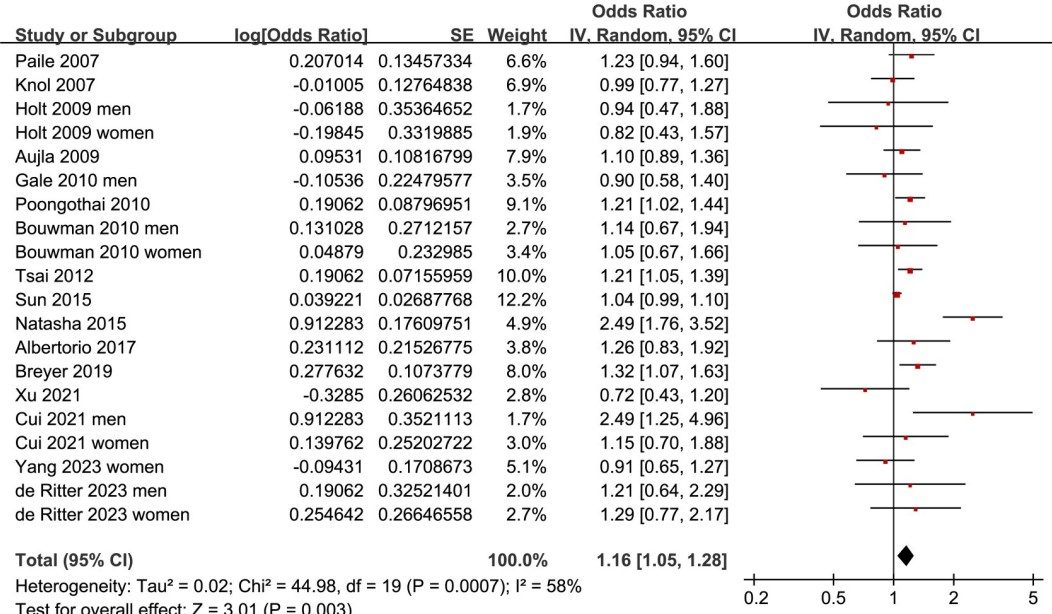

**Fig 2. Forest plots for the meta-analysis of the association between prediabetes and depression.**

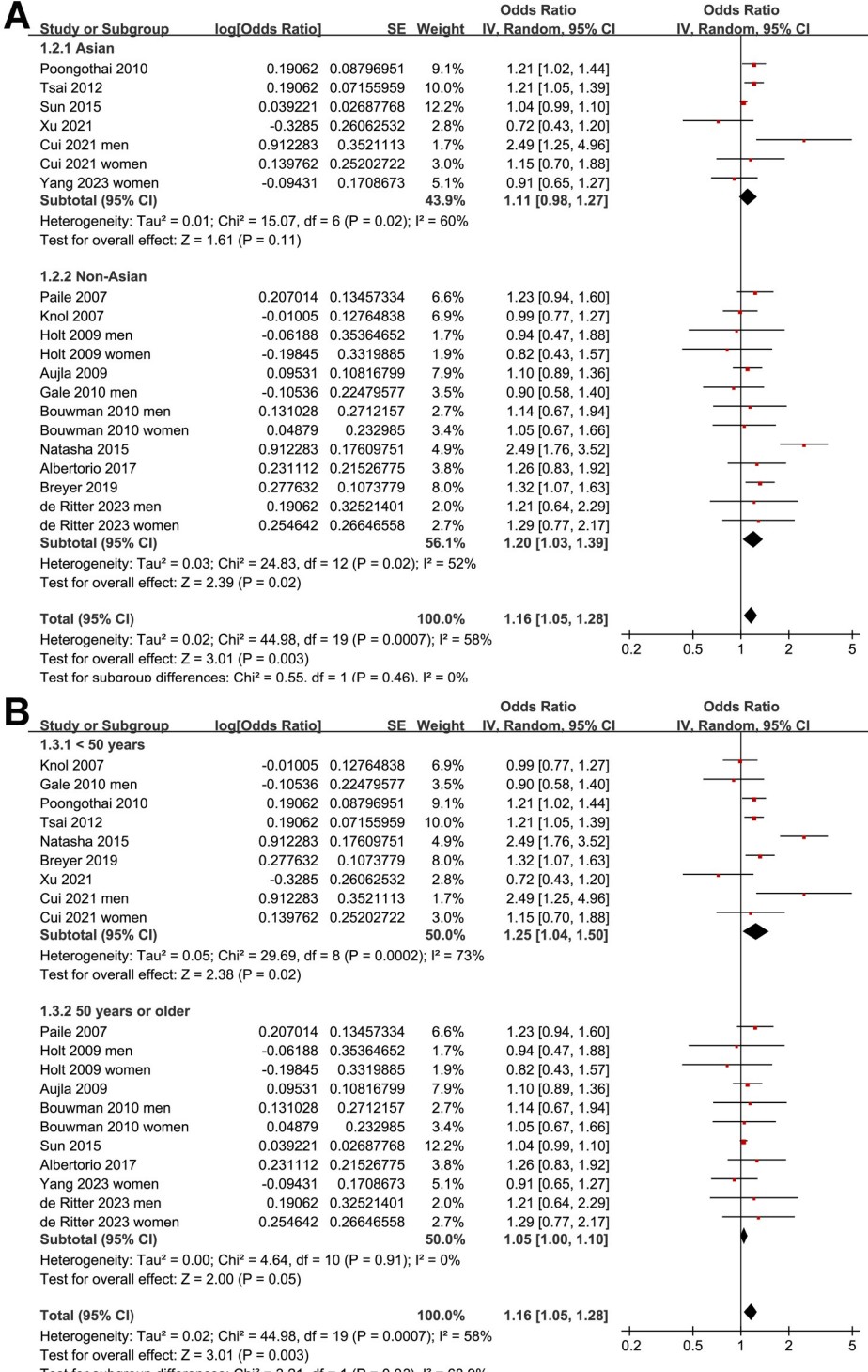

**Fig 3. Forest plots for the subgroup analyses of the association between prediabetes and depression: A, subgroup analysis according to the study country; and B, subgroup analysis according to the age of the subjects.**

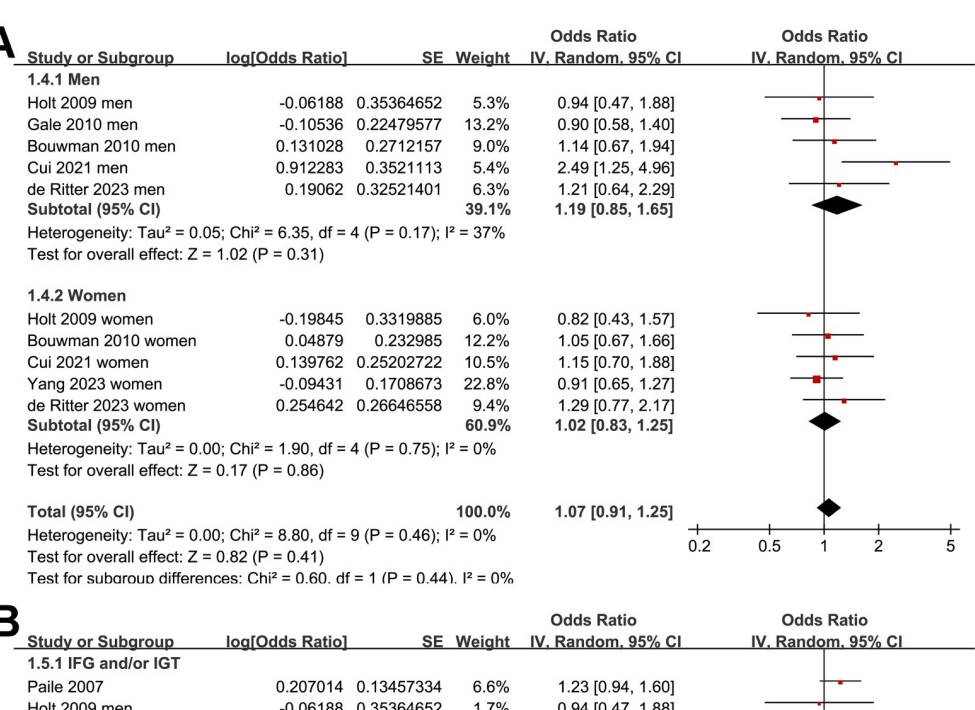

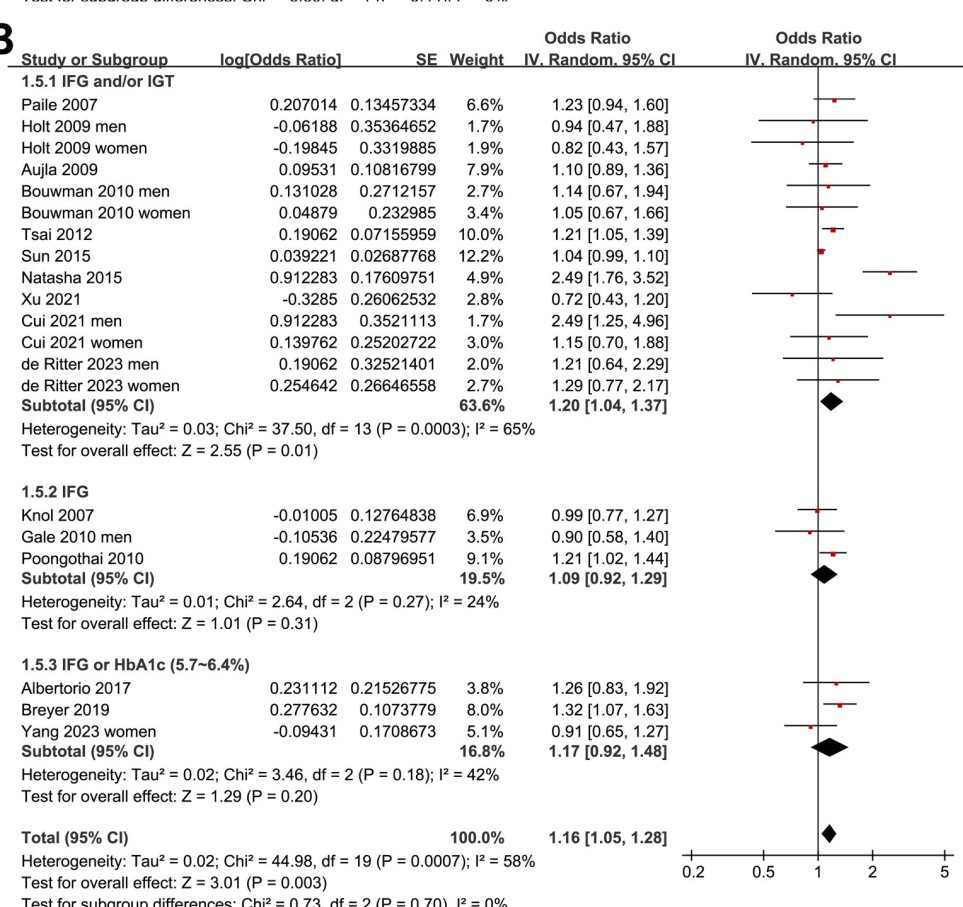

**Fig 4. Forest plots for the subgroup analyses of the association between prediabetes and depression: A, subgroup analysis according to the sex of the subjects; and B, subgroup analysis according to the definition of prediabetes.**

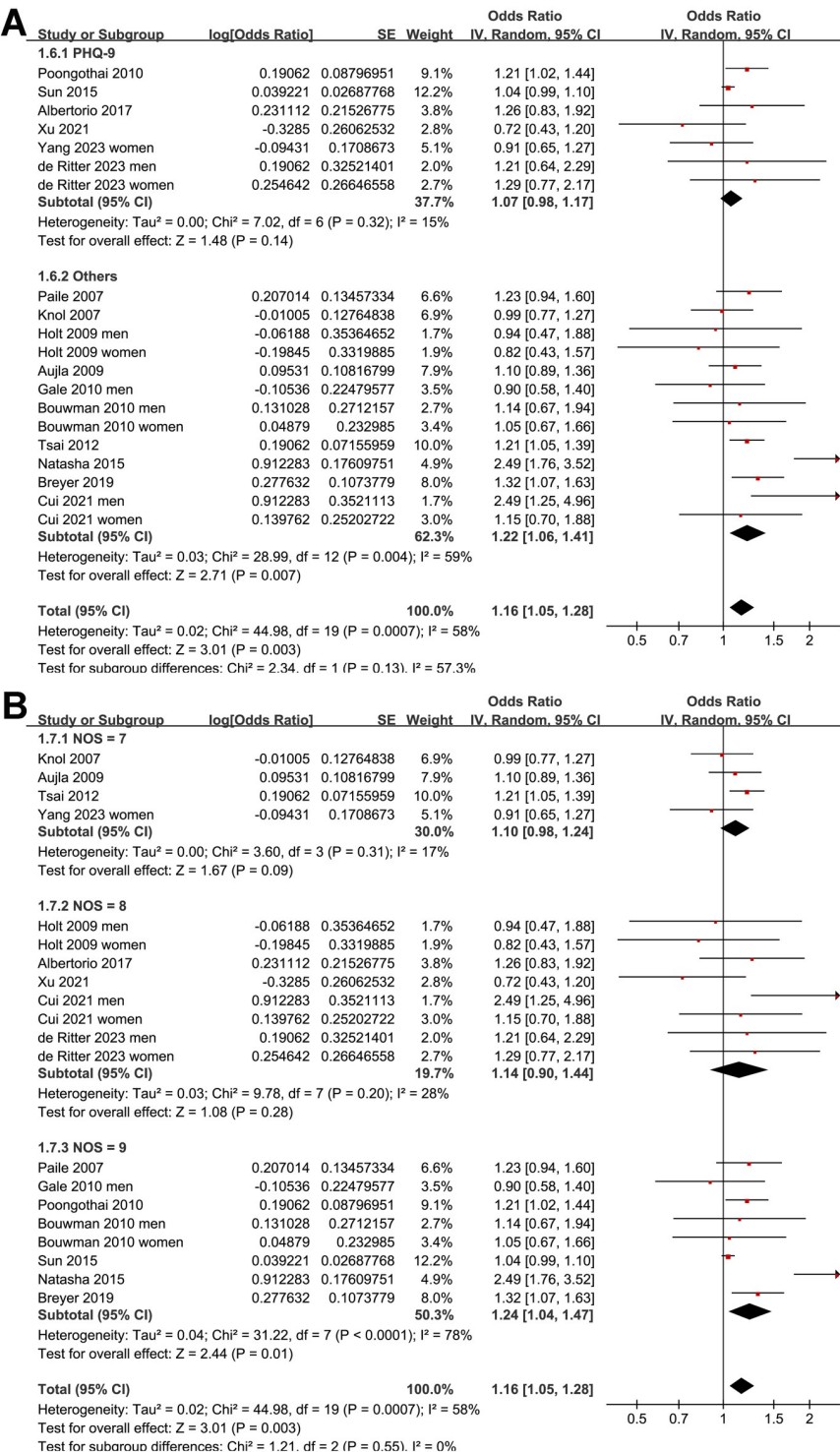

**Fig 5. Forest plots for the subgroup analyses of the association between prediabetes and depression: A, subgroup analysis according to the scales used for the diagnosis of depression; and B, subgroup analysis according to the study quality score.**

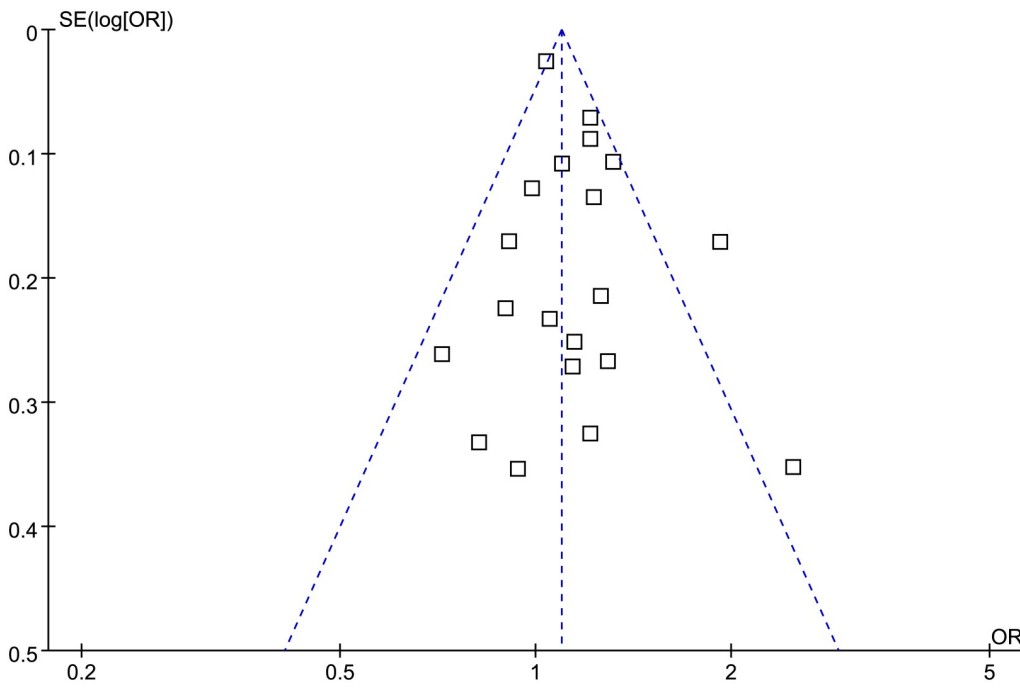

**Fig 6. Funnel plots for the meta-analysis of the association between prediabetes and depression.**

## Publication bias evaluation

The funnel plots for the meta-analysis of the association between prediabetes and depression were symmetrical upon visual inspection, indicating a low risk of publication bias (Fig 6). The results of Egger's regression test ($p = 0.55$) also suggested a low risk of publication bias.

## Discussion

In this systematic review and meta-analysis, we synthesized the evidence from 16 high-quality observational studies and found that compared to people with normoglycemia, those with prediabetes were associated with a slightly higher prevalence of depression. In addition, subgroup analysis according to age showed that the association between prediabetes and depression may be stronger in younger people <50 years old compared to older people aged over 50 years old. Subgroup analysis according to other study characteristics, such as the study country, sex of the participants, definition of prediabetes, methods used for the diagnosis of depression, and study quality scores, did not significantly change the results. Taken together, the results of the meta-analysis indicate that prediabetes may be associated a slightly higher prevalence of depression in the general population, particularly in subjects aged <50 years old.

To the best of our knowledge, few meta-analyses have been performed to evaluate the association between prediabetes and depression. Although, one early meta-analysis that included five observational studies suggested that impaired glucose metabolism was not associated with the development of depressive symptoms [17]. However, due to the limited number of available studies, no subgroup analysis could be performed. Compared to the previous meta-analysis, this current study has a few methodological strengths to highlight. First, an extensive literature search was performed in three commonly used electronic databases, which retrieved 16 large-scale high-quality studies according to the aim of the meta-analysis. We only included

large-scale studies with a sample size of at least 1000 to minimize the potential bias that can arise in small-scale studies. In addition, because depression is closely related to somatic diseases and multimorbidity [40], we focused on the general population, and excluded patients with specific diagnoses of diseases to avoid the confounding effects of comorbidities. Moreover, all of the included studies used multivariate analysis when the association between prediabetes and depression was determined, with adjustment for age, sex, and related socioeconomic status, which could minimize the confounding effects of these factors. Finally, multiple sensitivity and subgroup analyses were performed, and returned consistent results, further validating the robustness of the findings. Collectively, these results highlight the association between glycemic metabolism disorders in the development of depression symptoms, which may occur in prediabetes, even before the diagnosis of diabetes.

There are several hypotheses regarding the mechanisms underlying the association between prediabetes and depression. Persistent mildly hyperglycemia in prediabetes has been linked to chronic inflammation and oxidative stress [41], which has also been revealed in the pathogenesis of affective disorders, such as depression [42]. In addition, it has also been suggested that hyperglycemia and hyperinsulinemia in prediabetes could lead to neuroendocrine changes, which may finally stimulate the development of depression [43]. Moreover, similar to type 2 diabetes [44], prediabetes is also associated with cerebral microvascular dysfunction, which is also associated with a higher risk of depression [45]. However, it has to be mentioned that the above hypotheses have been rarely investigated in preclinical or clinical studies, and efforts are still needed to clarify the potential mechanisms underlying the association between prediabetes and depression.

The association between diabetes and depression is considered to be bidirectional [9–11]. Similarly, it is important to determine if the association between prediabetes and depression is also bidirectional. If the hypothesis is confirmed, this bidirectional relationship could suggest that there may be shared underlying mechanisms linking these two conditions, such as inflammation, hypothalamic-pituitary-adrenal axis dysregulation, and lifestyle factors. Furthermore, understanding this bidirectional relationship can inform future research directions and interventions aimed at preventing and managing both prediabetes and depression.

This study also has some limitations to note. First, all of the included studies were cross-sectional studies, which could not aid determining whether prediabetes is a risk factor for the development of depression. Prospective studies are needed in the future to address this issue. Secondly, the meta-analysis protocol was not registered in advance, which could affect the transparency of the methods. Thirdly, the definition of prediabetes and the diagnostic methods for depression varied among the included studies, which may be an important source of heterogeneity. However, as far as we know, the optimal definition of prediabetes and scale for evaluating depression remain to be established. Moreover, although the results were based on the data from multivariate analysis, we could not exclude the possibility that there are still unadjusted factors that may confound the association between prediabetes and depression, such as dietary and nutritional factors. For example, vitamin D deficiency has been related to both prediabetes [46] and depression [47], which therefore may confound the association between prediabetes and depression in the general population. In addition, other confounding factors such as family history, the presence of other comorbid conditions that might predispose to depression and life events need to be considered, especially given the cross-sectional nature of the studies included. However, since these factors were generally not reported in the included studies, we could not determine if they might have affected the results of the meta-analysis. Finally, the results of the meta-analysis showed that prediabetes is only associated with a mildly increased prevalence of depression, the clinical relevance of which remains to be determined.

## Conclusions

As a summary, results of the meta-analysis indicate that the prevalence of depression is slightly increased in people with prediabetes compared to those with normoglycemia, particularly in younger participants <50 years old. Large-scale prospective cohort studies are needed to determine if prediabetes is a risk factor for depression in general population.

## Acknowledgments

We thank Medjaden Inc. for scientific editing of this manuscript.

## Author Contributions

**Conceptualization:** Yi Yu, Weitao Wan.

**Data curation:** Yi Yu, Weitao Wan.

**Formal analysis:** Yi Yu, Weitao Wan.

**Investigation:** Yi Yu, Weitao Wan.

**Methodology:** Yi Yu, Weitao Wan.

**Project administration:** Weitao Wan.

**Software:** Yi Yu, Weitao Wan.

**Supervision:** Weitao Wan.

**Writing – original draft:** Yi Yu, Weitao Wan.

**Writing – review & editing:** Yi Yu, Weitao Wan.

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
