## [Decision Letter · Decision Letter 0]

22 Apr 2024

PONE-D-24-07500Association between prediabetes and depression: A meta-analysisPLOS ONE

Dear Dr. Wan,

Thank you for submitting your manuscript to PLOS ONE. After careful consideration, we feel that it has merit but does not fully meet PLOS ONE’s publication criteria as it currently stands. Therefore, we invite you to submit a revised version of the manuscript that addresses the points raised during the review process.

This study contributes to current knowledge about associations between depression and metabolic conditions, in this case, prediabetes. However, as the reviewers indicate clarity is needed on a number of issues including the role of confounding, the interpretation of the potential bidirectionality between prediabetes and depression and whether these findings support causality. A critical interpretation of the relevance of the selected studies to the outcomes would also be helpful, given that they take place in several populations and are cross sectional.

We look forward to receiving your revised manuscript.

Kind regards,

Anselm J. M. Hennis, MBBS, MSc, PhD, FRCP

Academic Editor

PLOS ONE

Journal Requirements:

https://www.frontiersin.org/journals/oncology/articles/10.3389/fonc.2023.1117846/full

In your revision ensure you cite all your sources (including your own works), and quote or rephrase any duplicated text outside the methods section. Further consideration is dependent on these concerns being addressed.

Reviewers' comments:

Reviewer's Responses to Questions

**Comments to the Author**

1. Is the manuscript technically sound, and do the data support the conclusions?

Reviewer #1: Partly

Reviewer #2: Yes

2. Has the statistical analysis been performed appropriately and rigorously? 

Reviewer #1: Yes

Reviewer #2: Yes

3. Have the authors made all data underlying the findings in their manuscript fully available?

Reviewer #1: No

Reviewer #2: No

4. Is the manuscript presented in an intelligible fashion and written in standard English?

Reviewer #1: Yes

Reviewer #2: Yes

5. Review Comments to the Author

Reviewer #1: The bases for conducting this review and meta-analysis are clear and solid. To better understand how previous studies on the relationship between prediabetes and depression are inconsistent, it is suggested to expand on the idea that they are inconsistent. Furthermore, considering that references 18-33 are studies with populations from various places and methodologies, this aspect could be briefly mentioned.

The objective of the review and meta-analysis is clear.

Methodological guidelines for review and meta-analysis are appropriate (PRISMA and Cochrane). It is striking that the manuscript does not detail whether the review protocol was published in Prospero or elsewhere. Presenting the protocol before data extraction promotes transparency of methods and reduces biases, which can be reviewed, and unnecessary duplication of effort between researchers.

Regarding the search terms, it is suggested to indicate whether they were compared with the MeSH terms.

Recommends that this entire paragraph be part of the “inclusion and exclusion criteria” (not the literature search): “the search was limited to studies in humans. Furthermore, we only considered studies published as full articles in peer-reviewed journals in English. As a complement, references of related original and review articles were manually examined to identify potentially related studies. Publications published from the inception of the databases to December 8, 2023 were examined.”

Given the results of the meta-analysis in the conclusions, it is suggested that prediabetes was associated with a slightly higher prevalence of depression.

The final statement of the conclusions is risky in light of these results since it suggests a causal relationship that the study has not determined. With these results, it is not possible to conclude that disorders of glycemic metabolism develop symptoms of depression.

Reviewer #2: Very useful study. Other confounding factorssuch as family history, the presence of other comorbid condirions that might predispose to depression and life events need to be considered. especially given the cross-sectional nature of the studies included. The time used may have been somewhat restrictive. In the introduction, the need to appreciate the bidirectional relationship of diabetes and depression and how this may have influenced the findings reported in the meta-analysis should also have been included.

6. PLOS authors have the option to publish the peer review history of their article (what does this mean?). If published, this will include your full peer review and any attached files.

Reviewer #1: **Yes: **Olga Toro-Devia

Reviewer #2: No

---

## [Author Response · Author response to Decision Letter 0]

7 May 2024

Dear Dr. Hennis and the reviewers of Plos One,

Thank you very much for your comments on our manuscript entitled “Association between prediabetes and depression: A meta-analysis” (PONE-D-24-07500). These comments are valuable for improving the quality of our work. We have revised the manuscript accordingly, with changes highlighted in red font. A detailed response letter has also been attached for your reference. Your further consideration is highly appreciated.

Look forward to hearing from you at your earliest convenience.

Best regards,

Corresponding author:

Weitao Wan

Department of Psychiatry, Tianyou Hospital Affiliated to Wuhan University of Science and Technology, No. 9, Tujialing, Dingziqiao Road, Wuchang District, Wuhan, Hubei, China. E-mail: wwt686@sohu.com

Reviewer #1:

(1) The bases for conducting this review and meta-analysis are clear and solid. To better understand how previous studies on the relationship between prediabetes and depression are inconsistent, it is suggested to expand on the idea that they are inconsistent. Furthermore, considering that references 18-33 are studies with populations from various places and methodologies, this aspect could be briefly mentioned.

Author’s reply: Thank you for your comments. We have briefly expanded the description of the inconsistent results of the previous studies as suggested in the revised Introduction part as “However, the results of previous studies were not consistent [18-33]. Some of them supported that prediabetes was related to depression [24-27, 29, 30], while other studies did not found a statistically significant association [18-23, 28, 31-33]. In addition, these studies are with populations from various places and of different study definitions and methodologies for evaluating prediabetes and depression [18-33]. It remains unknown whether these factors may influence the association between prediabetes and depression”.

(2) The objective of the review and meta-analysis is clear.

Author’s reply: Thank you for your comments.

(3) Methodological guidelines for review and meta-analysis are appropriate (PRISMA and Cochrane). It is striking that the manuscript does not detail whether the review protocol was published in Prospero or elsewhere. Presenting the protocol before data extraction promotes transparency of methods and reduces biases, which can be reviewed, and unnecessary duplication of effort between researchers.

Author’s reply: Thank you for your comments. The protocol of the manuscript was not registered prospectively. We have clarified this in the revised manuscript as a limitation of the study.

(4) Regarding the search terms, it is suggested to indicate whether they were compared with the MeSH terms.

Author’s reply: Thank you for your comments. The search terms were based on key words of rather than MeSH terms to improve the sensitivity of the database search. However, a comparison with MeSH terms was performed before database search to ensure all relevant MeSH terms are included in the search terms. This has been clarified in the revised Methods part. 

(5) Recommends that this entire paragraph be part of the “inclusion and exclusion criteria” (not the literature search): “the search was limited to studies in humans. Furthermore, we only considered studies published as full articles in peer-reviewed journals in English. As a complement, references of related original and review articles were manually examined to identify potentially related studies. Publications published from the inception of the databases to December 8, 2023 were examined.”

Author’s reply: Thank you for your comments. We have removed this paragraph from “literature search” to “inclusion and exclusion criteria” as requested.

(6) Given the results of the meta-analysis in the conclusions, it is suggested that prediabetes was associated with a slightly higher prevalence of depression.

Author’s reply: Thank you for your comments. We have revised the conclusions in the abstract and the manuscript as suggested to emphasize that prediabetes was associated with a slightly higher prevalence of depression.

(7) The final statement of the conclusions is risky in light of these results since it suggests a causal relationship that the study has not determined. With these results, it is not possible to conclude that disorders of glycemic metabolism develop symptoms of depression.

Author’s reply: Thank you for your comments. We have deleted this sentence accordingly. 

Reply to Reviewer #2

(1) Other confounding factors such as family history, the presence of other comorbid conditions that might predispose to depression and life events need to be considered, especially given the cross-sectional nature of the studies included.

Author’s reply: Thank you for your comments. We agreed with the reviewer on that other confounding factors such as family history, the presence of other comorbid conditions that might predispose to depression and life events need to be considered, especially given the cross-sectional nature of the studies included. However, since these factors were generally not reported in the included studies, we could not determine if they may affect the results of the meta-analysis. We have acknowledged this as a limitation of the study in the revised Discussion. 

(2) The time used may have been somewhat restrictive.

Author’s reply: Thank you for your comments. All studies published from database inception to the last search (December 8, 2023) were screened for possible relevance, which has been clarified in the methods part. With all respect, we believe the time used was not restrictive.

(2) In the introduction, the need to appreciate the bidirectional relationship of diabetes and depression and how this may have influenced the findings reported in the meta-analysis should also have been included.

Author’s reply: Thank you for your comments. We have mentioned in the introduction that “Interestingly, subsequent studies suggest that the association between diabetes and depression seems to be bidirectional [9-11]. Besides a high prevalence of depression in patients with diabetes, it is also shown that various measures of depression could be used to predict the risk of type 2 diabetes, such as depression as evidenced by symptom scales, patient diagnosis, face-to-face interviews, and the use of antidepressants [12]”. In addition, we have added some descriptions for the potential bidirectional relationship between prediabetes and depression in the revised Discussion part and to highlight the importance of this hypothesis on the interpretation of the meta-analysis, as “The association between diabetes and depression is considered to be bidirectional. Similarly, it is important to determine if the association between prediabetes and depression is also bidirectional. If the hypothesis is confirmed, this bidirectional relationship could suggest that there may be shared underlying mechanisms linking these two conditions, such as inflammation, hypothalamic-pituitary-adrenal axis dysregulation, and lifestyle factors. Furthermore, understanding this bidirectional relationship can inform future research directions and interventions aimed at preventing and managing both prediabetes and depression”.

---

## [Editor Report · Decision Letter 1]

11 Jun 2024

PONE-D-24-07500R1Association between prediabetes and depression: A meta-analysisPLOS ONE

Dear Dr. Wan,

Thank you for submitting your manuscript to PLOS ONE. After careful consideration, we feel that it has merit but does not fully meet PLOS ONE’s publication criteria as it currently stands. Therefore, we invite you to submit a revised version of the manuscript that addresses the points raised during the review process.

Thank you for reviewing and modifying the manuscript according to the recommendations of the reviewers. There are a few issues that still need to be addressed, chiefly grammatical in nature, and as follows:

line 93: ... search terms were based on key words [of - Delete please] rather than MESH terms

lines 100-101: Please amend text: studies conducted in adults (18 years and older)....

lines 114-115: As a supplementation - this is not grammatical - please amend..

line 121-123: and data collection were independently conducted by two authors. If disagreement occurred, a consultation with the corresponding author was carried out to resolve the disagreement. - There are in fact only two authors, and so following the decision of one author could potentially introduce biases - please clarify.

lines 277-278: Please correct 'second' to secondly;

The text 'Not registered prospectively which may influence the transparency of the Methods'.... is not grammatical and needs to be amended;

Please correct 'third' to thirdly.

line 292: please correct text to.... 'if they might have affected'

line 303: please correct to 'risk factor for depression'... ==============================

We look forward to receiving your revised manuscript.

Kind regards,

Anselm J. M. Hennis, MBBS, MSc, PhD, FRCP

Academic Editor

PLOS ONE
---

## [Author Response · Author response to Decision Letter 1]

20 Jun 2024

Dear Dr. Hennis and the reviewers of Plos One,

Thank you very much for your comments on our manuscript entitled “Association between prediabetes and depression: A meta-analysis” (PONE-D-24-07500R1). These comments are valuable for improving the quality of our work. We have revised the manuscript accordingly, with changes highlighted in red font. A detailed response letter has also been attached for your reference. Your further consideration is highly appreciated.

Look forward to hearing from you at your earliest convenience.

Best regards,

Corresponding author:

Weitao Wan

Department of Psychiatry, Tianyou Hospital Affiliated to Wuhan University of Science and Technology, No. 9, Tujialing, Dingziqiao Road, Wuchang District, Wuhan, Hubei, China. E-mail: wwt686@sohu.com

line 93: ... search terms were based on key words [of - Delete please] rather than MESH terms

Author’s reply: Thank you for your comments. This sentence has been revised accordingly.

lines 100-101: Please amend text: studies conducted in adults (18 years and older)....

Author’s reply: Thank you for your comments. This sentence has been revised accordingly.

lines 114-115: As a supplementation - this is not grammatical - please amend..

Author’s reply: Thank you for your comments. This sentence has been revised as “In addition, the references of related original and review articles were also manually screened for identifying potentially related studies”.

line 121-123: and data collection were independently conducted by two authors. If disagreement occurred, a consultation with the corresponding author was carried out to resolve the disagreement. - There are in fact only two authors, and so following the decision of one author could potentially introduce biases - please clarify.

Author’s reply: Thank you for your comments. We apologize for the inaccurate expression in this sentence. In case of disagreement, the two authors discussed it to reach a consensus. This has been revised in the manuscript.

lines 277-278: Please correct 'second' to secondly;

Author’s reply: Thank you for your comments. This sentence has been revised accordingly as “Secondly, the meta-analysis protocol was not registered in advance, which could affect the transparency of the methods”

The text 'Not registered prospectively which may influence the transparency of the Methods'.... is not grammatical and needs to be amended;

Author’s reply: Thank you for your comments. This sentence has been revised accordingly as “Secondly, the meta-analysis protocol was not registered in advance, which could affect the transparency of the methods”

Please correct 'third' to thirdly.

Author’s reply: Thank you for your comments. This sentence has been revised accordingly.

line 292: please correct text to.... 'if they might have affected'

Author’s reply: Thank you for your comments. This sentence has been revised accordingly.

line 303: please correct to 'risk factor for depression'...

Author’s reply: Thank you for your comments. This sentence has been revised accordingly.

---

## [Editor Report · Decision Letter 2]

5 Jul 2024

Association between prediabetes and depression: A meta-analysis

PONE-D-24-07500R2

Dear Dr. Wan,

We’re pleased to inform you that your manuscript has been judged scientifically suitable for publication and will be formally accepted for publication once it meets all outstanding technical requirements.

Kind regards,

Anselm J. M. Hennis, MBBS, MSc, PhD, FRCP

Academic Editor

PLOS ONE

Additional Editor Comments (optional):

Thank you for responding to the concerns raised.
---

## [Editor Report · Acceptance letter]

10 Jul 2024

PONE-D-24-07500R2 

PLOS ONE

Dear Dr. Wan, 

I'm pleased to inform you that your manuscript has been deemed suitable for publication in PLOS ONE. Congratulations! Your manuscript is now being handed over to our production team.

Kind regards, 

on behalf of

Dr. Anselm J. M. Hennis 

Academic Editor

PLOS ONE